# Stochastic co-teaching for robust cardiac segmentation in ultrasound with noisy labels

**Gino E. Jansen**[1]                                                    G.E.JANSEN@UVA.NL

**Mark J. Schuuring**[2]

**Berto J. Bouma**[3]

**Ivana Išgum**[1]

[1] *qurAI group, Informatics Institute, University of Amsterdam, Netherlands*

[2] *Department of Biomedical Signals and Systems, University of Twente, Netherlands*

[3] *Department of Cardiology, Amsterdam UMC UvA, Netherlands*

**Editors:** Accepted for publication at MIDL 2025

## Abstract

In this work, we propose a label noise-robust segmentation framework for left ventricle blood pool segmentation in echocardiography. Based on the stochastic co-teaching approach, our method extends pixel-level filtering of label noise with additional image-level filtering to more effectively prevent noisy labels from backpropagating. We evaluate our framework on the EchoNet-Dynamic dataset, and simulate diverse noisy label scenarios, including over- and undersegmented (i.e., biased) labels. Our results demonstrate that the incorporation of image-based rejection enhances the Dice coefficient by 1.5% points and ejection fraction estimation by 2.3% points with respect to the pixel-based co-teaching framework under heavily biased label noise conditions, and thereby maintains the same performance as on clean data.

**Keywords:** Echocardiography, Left Ventricle Segmentation, Deep Learning, Label Noise

## 1. Introduction

Transthoracic echocardiography (TTE) serves as the primary imaging modality for assessing cardiac function. A key quantitative parameter derived from TTE is the left ventricular ejection fraction (LVEF), which quantifies the ratio of blood ejected from the left ventricle relative to its total volume at end-diastole (Lang et al., 2015). Accurate automatic LVEF estimation depends on precisely delineated examples of the left-ventricle blood pool in TTE images, which may be compromised by less experienced annotators, who contribute noisy delineations (Karimi et al., 2020; Shi et al., 2024).

Various strategies have been developed to mitigate the adverse effects of label noise in deep learning. For example, Han et al. (2018) introduced the co-teaching framework, wherein two networks are trained simultaneously on the same mini-batches but cross-select different subsets of instances based on their loss values for optimization. In this approach, each network discards a predetermined fraction of samples with the highest losses to update its peer. Building on this idea, de Vos et al. (2023) proposed stochastic co-teaching (StoCoT), which employs randomly selected rejection thresholds, thereby improving performance in scenarios with unknown noise levels. The authors showed that co-teaching can easily be extended to segmentation tasks, as segmentation is usually framed as a pixel-wise classification problem, and single pixels can be masked out from backpropagation.

In this work, we employ the stochastic co-teaching framework for echocardiography left ventricle segmentation in echocardiography. To prevent a data imbalance between center and border pixels, we perform noisy-label rejection both on the pixel-level and on the image level, and show that this improves performance compared with stochastic co-teaching on the pixel-level only.

## 2. Method

We propose to adapt the stochastic co-teaching framework, which consists of two parallel networks that identify and filter out noisy labels for *each other's* optimization (cross-updates), thereby mitigating confirmation bias. In addition to the pixel-based rejection scheme proposed by de Vos et al. (2023), we add a frame-based rejection rule where entire images are excluded from training if they are deemed to contain excessive label noise.

The stochastic co-teaching algorithm operates as follows: During training, for each pixel, the posterior probability of the reference class is computed. For a pixel with binary label $y_{\text{true}} \in \{0, 1\}$, the network outputs a logit which is transformed via a sigmoid function to yield $y_{\text{pred}}$. The posterior probability is then defined as $P = y_{\text{true}} \cdot y_{\text{pred}} + (1 - y_{\text{true}}) \cdot (1 - y_{\text{pred}})$ for a binary classification task. A rejection threshold $\tau$ is sampled from a Beta distribution ($\alpha = 1$, $\beta = 8$); a pixel's loss is rejected from backpropagation through the network (i.e., masked out), if $P < \tau$ for the same pixel in the *peer* network. Additionally, an entire image is rejected, if more than 5% of its pixels are discarded through this process.

## 3. Experiments & Results

We evaluate our method on the EchoNet-Dynamic dataset (Ouyang et al., 2020), which consists of echocardiogram videos from 10,030 patients with expert delineations of the left-ventricle blood pool in end-systolic and end-diastolic frames. In addition to using the original annotations, we simulate label noise with three settings: **(a)** Positive bias 45%: 5 consecutive morphological dilation operations to 45% of the training images to mimic a consistently over-segmented blood pool by a contributing annotator. **(b)** Negative bias 45%: 5 consecutive erosions to 45% of the training images to simulate consistent under-segmentation. **(c)** Symmetric 80%: 1 to 5 erosions or dilations (uniformly random choice) to 80% of the training images to mimic inconsistent segmentation.

We compare our approach with four training strategies: **(i)** Baseline: A single model trained with binary cross-entropy loss from (Ouyang et al., 2020); **(ii)** co-teaching: Two networks that each mask out the 5% of pixels with the highest losses per image; **(iii)** stochastic co-teaching (pixel-based); and **(iv)** stochastic co-teaching (pixel- + image-based).

For all experiments, we adopt the settings from (Ouyang et al., 2020), including a DeepLab-v3 architecture with a ResNet-50 backbone, initialized with random weights, and weight optimization using SGD (learning rate 1e-5, momentum 0.9). Input images are resized to 112×112 and normalized to zero mean and unit variance. Training runs for 50 epochs, and we report results from the epoch with the best validation loss. Each experiment is repeated 5 times with different random seeds. The co-teaching experiments are performed with an initial warm-up phase of 10 epochs with regular training, and a subsequent 10 epochs of linear stepwise increase of the rejection threshold. Segmentation performance is assessed via the Dice coefficient on annotated frames and clinical importance is assessed using the absolute error of ejection fraction (EF). EF is computed using the end-systolic volume (ESV) and end-diastolic volume (EDV) according to: $\text{EF} = \frac{EDV - ESV}{EDV}$, with both volumes estimated as $\text{Volume} = 0.85 \times \frac{A^2}{L}$, where $A$ is the segmentation area, and $L$ the left-ventricle length (St John Sutton et al., 1998).

Table 1 summarizes the experimental results. The proposed image-based stochastic co-teaching framework outperforms all other methods in the positive bias 45% setting, achieving an average Dice score of 91.7%—comparable to performance on the clean dataset—and a mean absolute EF error of 9.0 percentage points. Under the Symmetric 80% noise condition, the framework yields a modest Dice improvement, though with a slight decline in EF accuracy, while no significant gains are observed for the Negative bias scenario. The experiments on the clean dataset show that the application of stochastic co-teaching preserves baseline performance. Figure 1(b) illustrates

Table 1: Results on the test set, reported as mean ± standard deviation (computed across 5 random seeds). Dice scores are in % and MAE of EF in percentage points. Best means are in bold; * indicates a significant improvement ($p < 0.05$, Bonferroni-corrected, paired t-tests).

| | Clean | | Pos. bias 45% | | Neg. bias 45% | | Sym. 80% | |
|---|---|---|---|---|---|---|---|---|
| Method | Dice | MAE | Dice | MAE | Dice | MAE | Dice | MAE |
| Baseline | 91.7 ± 0.3 | **8.6 ± 0.3** | 89.7 ± 0.6 | 12.8 ± 2.0 | **90.0 ± 0.6** | 10.7 ± 1.5 | 90.2 ± 0.4 | **9.7 ± 0.3** |
| Co-teaching | 91.3 ± 0.3 | 9.5 ± 0.5 | 89.4 ± 0.9 | 12.5 ± 1.6 | 88.8 ± 1.0 | 10.7 ± 2.1 | 88.4 ± 4.4 | 13.1 ± 5.8 |
| StoCoT (pixel) | **91.8 ± 0.1** | 8.7 ± 0.3 | 90.2 ± 0.4 | 11.3 ± 1.6 | 89.4 ± 0.6 | 11.1 ± 1.2 | 90.6 ± 0.5 | 10.1 ± 1.1 |
| StoCoT (pixel+image) | **91.8 ± 0.2** | **8.6 ± 0.4** | **91.7 ± 0.2*** | **9.0 ± 0.4** | 88.5 ± 0.3 | **9.9 ± 1.1** | **90.9 ± 0.3** | 10.9 ± 0.8 |

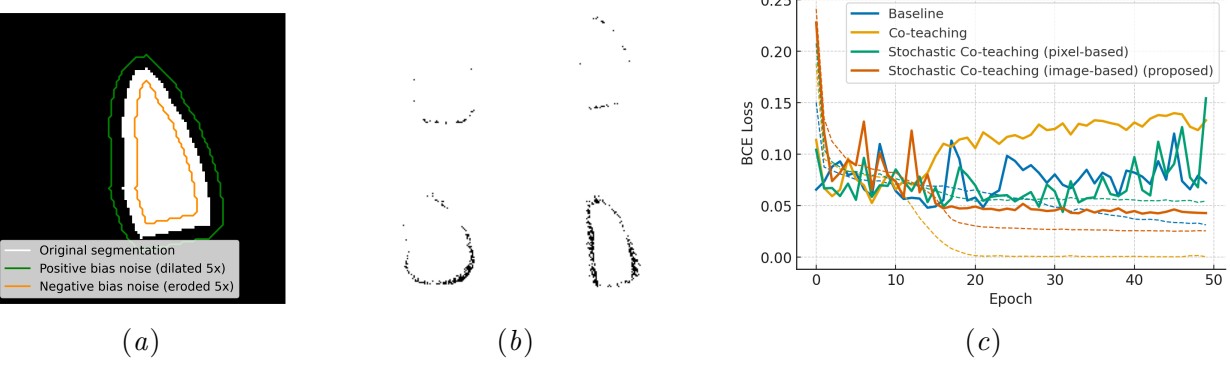

$(a)$             $(b)$             $(c)$

Figure 1: (a) Synthetic label noise examples. (b) Selection masks from stochastic co-teaching (black pixels: rejected; top two are clean, bottom two noisy). (c) Loss curves (solid: validation, dashed: training).

example pixel rejection masks generated during training, where noisier segmentation labels lead to broader rejected regions around the borders. Furthermore, as shown in Figure 1$(c)$, all methods except the proposed image-based approach exhibit signs of overfitting.

## 4. Discussion and conclusion

We applied and evaluated stochastic co-teaching for training models to segment the left ventricle blood pool in echocardiography. Our method extends the stochastic co-teaching framework by incorporating image-level rejection in addition to pixel-level rejection. This dual rejection strategy effectively filters out mislabeled data, as evidenced by improved performance compared to using pixel-based rejection alone.

We hypothesize that relying solely on pixel-based rejection may lead to an imbalance in training: excessive rejection of border pixels could result in a predominance of central pixels during backpropagation, thereby impairing the learning of boundary details. The image-based rejection helps mitigate this imbalance, ensuring that both border and central regions contribute adequately to the training process. In the negative bias experiment, the proposed method failed because it treated the clean labels as the noisy ones, and vice versa, indicating that a 45% noise rate was too high for this setting. However, in a more realistic noise scenario, the noisy annotator may also be less consistent. Future work should evaluate this method on label noise from expert annotators, and investigate if and why the method underperforms in scenarios with negatively biased noise.

## Acknowledgments

This work was supported by Esaote SpA and Pie Medical Imaging BV.

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
