# OpenReview forum: "Stochastic co-teaching for robust cardiac segmentation in ultrasound with noisy labels"
_MIDL.io/2025/Short_Papers — MIDL 2025 - Short Papers_

### Official Review · Reviewer_tasA · 2025-04-28

**Rating:** 4
**Confidence:** 4

**Summary:**

This paper presents a label noise-robust segmentation framework for left ventricle blood pool segmentation in echocardiography.
The authors adapt the stochastic co-teaching framework,by adding a frame-based rejection rule where entire images are excluded from training if they are deemed to contain excessive label noise. The method is evaluated on the EchoNet-Dynamic dataset (echocardiogram videos from 10,030 patients with expert delineations of the left-ventricle blood pool in end-systolic and end-diastolic frames).

**Strengths:**

This paper is clear and well-written.
Several experiments with different strategies have been carried out  on the EchoNet-Dynamic dataset.

**Weaknesses:**

No specific weakness for a short paper. More experiments could be done on different datasets.

---

### Decision · Program_Chairs · 2025-05-01

Accept